# Advances of Quantum Key Distribution and Network Nonlocality

**DOI:** 10.3390/e27090950

**Published:** 2025-09-13

**Authors:** Minming Geng

**Affiliations:** 1School of Computer, Electronics and Information, Guangxi University, Nanning 530004, China; gengmm@gxu.edu.cn; 2Guangxi Key Laboratory of Multimedia Communications and Network Technology, School of Computer, Electronic and Information, Guangxi University, Nanning 530004, China

**Keywords:** quantum key distribution, Bell nonlocality, Bell inequality, network nonlocality, full network nonlocality, genuine network nonlocality

## Abstract

In recent years, quantum network technology has been rapidly developing, with new theories, solutions, and protocols constantly emerging. The breakthrough experiments and achievements are impressive, such as the construction and operation of ultra-long-distance and multi-user quantum key distribution (QKD) networks, the proposal, verification, and experimental demonstration of new network nonlocality characteristics, etc. The results of recent research on QKD and network nonlocality are summarized and analyzed in this paper, including CV-MDI-QKD (continuous-variable measurement-device-independent QKD), TF-QKD (twin-field QKD), AMDI-QKD (asynchronous MDI-QKD), the generalization, sharing, and certification of network nonlocality, as well as the main achievements and related research tools of full network nonlocality and genuine network nonlocality, aiming to identify the current status and future development paths of the QKD and network nonlocality.

## 1. Introduction

In 1964, Bell derived Bell’s inequality [1] based on the local realism and implicit variable hypothesis [2] proposed by Einstein et al. and the nonlocal phenomena [3] observed in the microscopic world described by quantum theory, proving the existence of quantum nonlocality and indicating the experimental conditions for verifying nonlocality, which effectively promoted the development of quantum theory and experiments. The nonlocality of quantum mechanics can be verified by witnessing the violation of Bell’s inequality, which was first demonstrated in 1972 [4] with loopholes. In 1982, the experiment led by Aspect and his group closed the localized loopholes and confirmed the quantum nonlocality [5], which paved the way for the development of the quantum field. Quantum technologies have flourished in recent years, such as quantum computing [6,7,8], quantum communications [9,10,11], quantum sensing and measurement [12,13,14], and so on. In this paper, the advance of quantum network nonlocality in recent years will be reviewed.

Quantum and network are seemingly two unrelated ideas, but when they are combined together, they have been attracting widespread research attention, achieving fruitful results [15,16], and providing hot topics to be studied continuously. At the beginning, the concept of quantum networks was used in both the communication and computing fields. The gate arrays used in the quantum computing systems were once called quantum networks [17,18], which was used to describe the complexity and irregularity of the connections in the gate arrays. In 1992, the first quantum key distribution (QKD) experiment [19] was carried out successfully by Bennett et al., which made the rapid development of quantum communication technology. Although the connotations of classical communication and quantum communication are quite different, they can share the same communication medium, like satellite [20], optical fiber [21], free space [22], and so on. Therefore, it is natural for quantum technology to share the network with classical technology [23], and quantum networks are gradually becoming exclusively used in the field of quantum communication.

The basis of quantum network nonlocality is Bell nonlocality. Therefore, the characteristics of standard Bell nonlocality are briefly introduced in Section 2. The progress of quantum key distribution in recent years is summarized in Section 3. The integration of Bell’s theory and networks has introduced new connotations to nonlocality. The progress of research on network nonlocality is summarized in Section 4. With the vigorous growth of quantum communication technology, quantum networks based on traditional Bell nonlocality have been limited in both scale and structure, which cannot effectively solve the problems originating from complex networks. In 2022, the idea of full network nonlocality [24] was proposed by A. Pozas Kerstjens et al., which is a powerful tool for breaking through the limitations of classical correlation frameworks and will be reviewed in Section 4.4. In Section 4.5, the progress of genuine network nonlocality correlated with device-independent characteristics is reviewed.

## 2. Bell Nonlocality

Bell nonlocality was proposed by John Bell [1] to describe the non-classical correlations between two or more spatially separated quantum systems, which is quantified through Bell inequality (shown in Equation (1)), indicating that the measurement results of these systems cannot be explained by classical local hidden variable (LHV) theory [25,26,27].(1)$$1+p(x,y)\geq \left|p\left(x,z\right)-p(y,z)\right|$$(2)$$p\left(\alpha ,\beta \right)=\int_{\Omega }\rho \left(\lambda \right)A(\alpha ,\lambda )B(\beta ,\lambda )d\lambda$$

Here, *x* ∈ X, *y* ∈ Y and *z* ∈ Z are measured by different observers, often named Alice, Bob and Charlie, in a Bell test. Alice and Bob obtain *α* ∈ A and *β* ∈ B through Bell state measurement (BSM), respectively. The experimentally accessible probability is the joint probabilities denoted by p(*x*, *y*), p(*x*, *z*), and p(*y*, *z*), respectively. A hidden variable *λ* is introduced in the LHV theory, and the probability distribution of *λ* satisfies *ρ*(*λ*) ≥ 0, $\int_{\Omega }\rho (\lambda )d\lambda =1$, *λ* ∈ Ω. To simplify the expression of Equation (2), p(*α*, *β*|*x*, *y*) is denoted by p(*α*, *β*), while p(*α*|*x*, *λ*) and p(*β*|*y*, *λ*) are denoted by A(*α*, * λ*) and B(*β*, *λ*) with the assumptions of the locality condition and the measurement independence [25].

Thus, the violation of Bell’s inequality proves the nonlocality of quantum mechanics. In order to verify the nonlocality of quantum states, different types of Bell inequalities [28,29,30], Hardy paradoxes [31,32,33], GHZ paradoxes [34,35], etc., have been proposed and demonstrated by experiments [36,37,38]. Though several significant loophole-free experiments have been verified [39,40,41], this issue is still being discussed in depth by the academic community [26,27,42,43,44,45]. In 2024, Aiello proposed a new inequality-free method that verifies Bell nonlocality by comparing correlation functions instead of using inequalities, avoiding the counterfactual reasoning loopholes in traditional experiments and simplifying the falsification process of LHV theory. The joint probability distribution in Equation (2) can be expressed by the Fourier series of periodic variables in Equation (3) [46].(3)$$p\left(x,y\right)\overset{\begin{array}{c} A\left(x+\pi ,\lambda \right)=A\left(x,\;\lambda \right)\\ B\left(y+\pi ,\lambda \right)=B\left(y,\;\lambda \right) \end{array}}{\to }\int_{\Omega }\rho \left(\lambda \right)F(x,\lambda )F(y,\lambda )d\lambda$$

Here, *x* as well as *y* is the angle of the polarization analyzer with a period of π, which is common equipment in the BSM. The function *F* can be expanded using Fourier series as Equation (4).(4)$$F\left(\theta ,\lambda \right)=\sum_{n=-\infty }^{+\infty }f_{n}(\lambda )\frac{e^{2jn\theta }}{\sqrt{\pi }}$$

Here, *f_n_*(*λ*) are Fourier series coefficients dependent on *λ*.

Fourier series expansion is used as the main tool to represent the correlation functions predicted by quantum mechanics and LHV theory, and the conflict between LHV and quantum mechanics is proved through the inconsistency of their Fourier series coefficients, which provides a more intuitive and logically clear scheme to test the conflict between LHV and quantum mechanics. The method does not require simultaneous measurement of incompatible observations, making it easier to achieve in experiment, and indicates that BSM with periodic variables [47,48] can use methods similar to Fourier series expansion [49,50,51,52] to simplify the complexity in measuring Bell-type inequalities. The Ramsey and Hahn echo experiments are used to measure the decoherence properties of Bell states [47], where Ramsey interference experiments typically introduce periodic signals (such as radio frequency pulse sequences), and the periodic adjustment of their time parameters implies the use of periodic variables. The Mach–Zehnder interferometer is used to realize Wheeler’s delayed-choice experiment [48], and the interference phenomenon itself is a periodic manifestation.

## 3. Quantum Key Distribution

QKD is an important application of Bell’s nonlocality in network encryption transmission, which promises theoretic security based on the quantum theory [53,54,55,56,57,58]. According to the physical properties of quantum state carriers, there are mainly two types of QKD schemes, named discrete variable QKD (DV-QKD) [59] and continuous variable QKD (CV-QKD) [60], respectively. DV-QKD is based on single-photon technology and protocols of BB84 [61] and B92 [62], which support long-distance communications. CV-QKD is based on the continuous regular component of the light field with a balanced detector in the receiver, which supports high-speed communications. In this section, several mainstream QKD schemes will be summarized and discussed.

### 3.1. CV-MDI-QKD

To remove detector side-channel attacks, the measurement device-independent QKD (MDI-QKD) scheme was proposed in 2012 [63], which was subsequently subject to extensive research and development. Continuous-variable MDI-QKD (CV-MDI-QKD) [64] is preferred to discrete-variable MDI-QKD (DV-MDI-QKD) [65] due to the relatively low difficulty in implementation and efficient measurements. The first CV-MDI-QKD system built on optical fiber over 10 km, which can provide a security key rate (SKR) of about 0.19 bit/pulse, was reported in 2022 [66]. To raise the security key bitrate, the CV-MDI-QKD systems with 10 GBaud and 8 GBaud symbol rates were demonstrated [67], which can realize an SKR larger than 700 Mbit/s with high-order quadrature amplitude modulation (QAM) over a 5 km optical fiber. The received signals are processed by an integrated receiver (shown in Figure 1a) and offline DSP, which employs an electric pilot to estimate the frequency and phase offsets between the transmitter and receiver. Integrated CV-MDI-QKD can minimize the volume of the system and provide a solution to construct low-cost, highly stable, and robust quantum communication networks. An integrated CV-QKD system with an on-chip source, which is shown in Figure 1b, is demonstrated [68] and provides an SKR of about 900 bit/s with the optical fiber about 100 km. After that, the loopholes of the integrated light sources are analyzed in the CV-MDI-QKD system [69], such as the noise leakage, spectrum attack, purity attack, and so on, and various countermeasures are proposed at the same time. The integrated solutions for sending and receiving have been implemented, making it possible to realize a fully integrated CV-MDI-QKD system. Of course, a comprehensive analysis of the potential loopholes of the system is required.

Optical pilot is a commonly used scheme for the synchronization of frequency and phase between the distant sources but usually requires an additional fiber channel to realize the optical phase-locked loop (OPLL) [66], which is shown in Figure 2a. Coarse wavelength division multiplexing technology has been adopted [70], which multiplexes the reference light and quantum signals in a single optical fiber and simplifies the QKD system (shown in Figure 2b). Considering the problems faced by QKD systems in star network applications, the asymmetric CV-MDI-QKD protocol was adopted, which realized an SKR of about 2.6 Mbit/s with a 20 MBaud symbol rate over 10 km of fiber. The SKR can be increased efficiently by adopting high-order modulation and a higher symbol rate [67].

### 3.2. TF-QKD

Twin-field QKD (TF-QKD) was proposed in 2018 [71] to break through the linear boundary limitation of the repeaterless QKD system [72] and extend the transmission distance of MDI-QKD systems beyond 500 km. The maximum transmission distance of the TF-QKD reached 833.8 km in 2022 [73] and exceeded 1000 km in 2023 [74]. In order to overcome the channel distortion of ultra-long optical fiber, various techniques were adopted to improve the SKR and accuracy of the system [74], such as the sending-or-not-sending protocol [75], the three-intensity decoy-state method [76], frequency stabilization technology [77], and the actively odd-parity-pairing (AOPP) method [78]. MDI-QKD systems typically require optical reference light [67,70] or optical phase-locking technology [73,74] to monitor and compensate for frequency and phase offsets between remote lasers, which will increase the complexity and noise of the system.

Several lightweight MDI-QKD systems have been demonstrated experimentally [79,80,81,82]. The local optical frequency comb (OFC) measurement method has been proposed [79] to dismiss the fiber channel used for optical frequency dissemination [74]. A field test in Shandong, China, between Jinan and Qingdao was carried out based on the OFC-TF-QKD scheme [80] over 546 km of optical fiber, which was verified to support the asymmetric channels (about 44 km of fiber length difference) and ease the deployment of the system. Another field test in Germany was carried out between Kehl and Frankfurt over a 254 km deployed commercial network [81]. The reference light generated by the central node (Charlie) was distributed to the communication nodes (Alice and Bob) through quantum mux/demux, and optical injection locking (OIL) technology was used to achieve the frequency locking of the light sources. In order to maintain the coherence between signals in long-distance coherent quantum communication, off-band phase stabilization technology was proposed, which uses two optical signals with similar but different frequencies to suppress channel phase disturbance. One of the optical signals acts as a pilot (reference channel) and carries channel disturbance information after transmission. The channel disturbance information is used to compensate for another optical channel (quantum channel) and achieve coarse phase compensation. Replacing the superconductive nanowire single photon detectors (SNSPD) with avalanche diodes (APD) operating at −30 °C brings quantum communication systems closer to classical communication systems. A time division multiplexing (TDM) scheme with a fast Fourier transform-based algorithm is proposed to track the frequency and phase fluctuation in the TF-QKD system without an additional phase-locking fiber channel [82], and a post-processing method is used to reconcile the phase in the reference frame. The ratio between the reference frame and the quantum frame may influence the synchronization performance and the SKR of the system.

### 3.3. AMDI-QKD

Most MDI-QKD systems operate in synchronous mode, requiring precise phase tracking and compensation throughout the entire network, which is difficult to achieve as the scale of the quantum network increases. In the face of the issue, asynchronous MDI-QKD (AMDI-QKD) has been proposed [83,84], demonstrated experimentally [85,86], and optimized [87,88]. In [86,89], separate lasers with an Allen deviation frequency better than 10^−10^ could be used to implement the twin-field QKD (TF-QKD) systems, which eliminates the frequency locking channel and improves the robustness of the systems. The frequency stabilization method based on acetylene saturation absorption spectroscopy not only successfully meets these stringent performance requirements (especially achieving extremely high frequency stability and minimal interlaser frequency difference) but also avoids complex frequency locking techniques, such as complex optical phase-locked loop [66] or heterodyne locking [90]. A field test in Zhejiang, China, with a symmetric link between Yiwu and Lishui and an asymmetric link between Jinhua and Lishui, was carried out in 2024 [91], which is shown in Figure 3a. A strong light was used as a pilot and transmitted in the same fiber channel as the quantum signal in the form of TDM, which was used to estimate the frequency deviation introduced by lasers and optical fiber. A strong reference light may introduce noise, which can be suppressed by a narrow band optical filter. But the quantum transmission efficiency is inevitably affected by the TDM scheme, which will shrink the effective SKR of the QKD system. Thanks to the technology of post-measurement coincidence pairing, only the published quantum signal detection results are used in the AMDI-QKD system to estimate the frequency offset between distant lasers, which eliminates the dependence of the QKD on strong reference light and promotes the quantum transmission efficiency, making full use of the duty cycle [92]. The experimental scheme is shown in Figure 3b. By utilizing ultra-stable sources, the SKR of 1.03 Mbit/s (150 bit/s) over a fiber channel of 100 km (504 km) can be achieved.

Several recent experimental results of MDI-QKD systems are listed in Table 1. The integration solutions for CV-QKD tend to mature, which can effectively lower the difficulties in deployment, operation, and maintenance of the systems. And the SKR of CV-QKD can reach and exceed 1 Gbit/s by utilizing higher-order modulation patterns and higher clock rates. The transmission distance and SKR of the TF-QKD can be further improved by adopting super phase stabilization technology [93] to compensate for the phase fluctuations induced by optical fiber, an integrated solution [94,95,96] to reduce losses at the transmitter and receiver, and fast real-time dynamic calibration technology [97] to improve system stability. To lower the complexity and promote the performance of the QKD systems, new hybrid schemes have been proposed, such as the RFI-AMDI-QKD [87], QKD-KEM protocol [98], PQC-QKD protocol [99], hybrid encoder [100], photonics-atomic system [101], hybrid entangled photon pairs [102], and so on.

### 3.4. Discussion

In MDI-QKD networks, the assumption of an independent source is one of the core prerequisites for ensuring system security. If the light sources are not independent, side channel attacks [103,104,105,106,107] may be carried out by manipulating the correlation between different sources, which can compromise the security of quantum keys. But at the same time, the independent-source assumption is also a major factor limiting the deployment and performance of QKD systems. In the face of imperfect devices, complex network structures, and different user requirements, MDI-QKD schemes with asymmetric channels and non-ideal light sources have been proposed and optimized [108,109,110]. The analysis yield bounds are derived when different decoy intensity settings are used by the parties, and the influence of the asymmetric channel and the intensity fluctuations of the sources are considered to optimize the TF-QKD protocol [108]. An SNS-TF-QKD protocol based on an asymmetric source was proposed, and the security of the protocol is guaranteed by the mathematical constraints of the source parameters and proved by three virtual protocols and reductions [109], which enhances the practicality and flexibility of the TF-QKD. Based on it, the source parameters were optimized to increase the SKR of the asymmetric TF-QKD systems. The asymmetric TF-QKD protocol was improved by the analysis of the intensity fluctuation of the unstable sources and the statistical fluctuation of the quantum data sizes [110], and the security of the system was verified through simulation. A full asymmetric MDI-QKD scheme has been proposed [111], and the interference phenomenon of multiple photons on a beam splitter under different temporal modes, including polarization information, is investigated in this paper, and more details of asymmetric source communication in different modes of MDI-QKD are studied. The key rates of MDI-QKD systems using weak coherent pulse sources [112,113] and spontaneous parametric down-conversion sources [114,115] were modeled and simulated in three different scenarios (two symmetric and one asymmetric). The simulation results indicate that the SKR with asymmetric sources is always comparable to that with symmetric sources throughout the entire transmission distance, which means the limitations of the MDI-QKD systems can be simplified, and a practical solution for the asymmetric scenarios that are inevitably encountered in practical applications is proposed.

Temporal mode matching [111] is the theoretical basis for analyzing and optimizing the performance of MDI-QKD systems, and temporal photon interference processes with different modes and quantities have been analyzed [116,117,118,119]. A theoretical framework of stimulated Raman emissions for various physical systems has been established [116], which provides a theoretical basis for qubit shaping in different physical systems, shows how to improve the fidelity by optimizing the pulse shape, and provides a paradigm shift from drive optimization [120] to temporal-mode optimization with a fixed drive. Temporal mode matching has been used to describe the relationship between the dynamics of the quantum emitter and the shape of the flying qubit [116]. By solving the time-dependent coupling strength in the input-output theory, which is equivalent to temporal filtering, the conditions for efficient high-fidelity transfer of quantum states can be found. The vacuum state has been used to construct the virtual cavity in the input-output theory [121,122]. However, the vacuum state is not a logical state, which cannot support the error detection in the heralding protocol [117,118]. Thus, the vacuum state in the three-level system is replaced by the ground state, which is denoted by ${|\left.g\right\rangle }_{D}$ in Figure 4b. The temporal modes processed by two shaping filters (as shown in Figure 4a), which are defined as normalized boxcar functions, are orthogonal and overlap in time, allowing them to be independently physically extracted and mapped to different quantum memories. The selected modes will entangle in the frequency range of (*ω*_1_
*– ω*_2_) = 2 *m*π/*T*, where *m* is an arbitrary integer and *T* is the duration of the filter functions.

By applying this scheme in superconducting circuits, the matching hybridization of symmetric and anti-symmetric modes is achieved, and the required photon temporal modes are extracted from the waveguide (as shown in Figure 4b), achieving frequency-bin photon encoding for the first time in the microwave frequency band. The frequency-encoded photons can relieve the impact of birefringence on entanglement fidelity in quantum networks by replacing polarization-encoded photons [119,123]. The birefringence of the optical fiber can be eliminated by using polarization-maintaining fiber, while that of the cavity is inevitable. The frequency states of the photons will not be affected by time-dependent polarization oscillations in the cavities, which can maintain high stability in the transmission of quantum networks [119]. And the influence of the cavity birefringence can be further mitigated by improving the temporal mode matching between the networking nodes, such as the detuning of the lasers, the Rabi oscillation of the lasers, the coupling strength between the atom and the cavity, and so on. The theoretical framework for atomic cavity quantum networks was established and the generation process of two-phonon interference entanglement was analyzed in depth. Two feasible experiment schemes were proposed, which used the atomic fine-structure splitting of ^40^Ca^+^ and the hyperfine splitting of ^225^Ra^+^ ions, respectively. Following this method, ^11^B^3+^ ions [124], as well as ^87^Rb, ^211^Fr and ^225^Rb^+^ ions [125], can be used as the cavity.

## 4. Network Nonlocality

QKD is a secure key sharing technology based on the principles of quantum mechanics, while network nonlocality refers to the non-classical correlation exhibited by quantum states in distributed networks, which cannot be reproduced through LHV theory. The intrinsic relationship between QKD and network nonlocality is mainly reflected in the fact that the network nonlocality is the core foundation and efficiency improvement resource for QKD security. The security of QKD relies on network nonlocality, especially device-independent QKD (DI-QKD), which verifies device security by violating Bell’s inequality [126]. Nonlocality of the network serves as a quantifiable resource to guide the architecture design of QKD networks [127] and improve the efficiency of QKD [128].

After years of construction, many countries have established QKD networks for theory and application research [74,80,91,129,130,131,132,133,134,135]. And to the best of our knowledge, the quantum network between Beijing and Shanghai is the largest long-distance QKD system built at present [134], which is composed of multiple metropolitan area networks and more than 50 relays and integrates satellite links to realize 4600 km of quantum communication. However, there are still some problems to be solved in QKD networks, such as the vulnerability of qubits to environmental interference [107,136,137,138,139,140], which makes QKD less compatible with existing optical fiber networks, the slow standardization process [141,142,143], and increasingly serious information security problems [7,103,104,105,106,107,144,145]. The verification of network nonlocality can prove the independence and non-classical distribution of entanglement sources in quantum networks [62,146,147,148,149,150,151,152,153,154,155,156,157,158,159], which can avoid the potential risk of deceptive attack [160,161,162] from intermediate nodes in traditional communication systems. Furthermore, different network topologies can introduce correlations beyond the classical limitation [163,164,165,166,167], revealing the potential of high-dimensional entanglement in resisting channel distortion and improving SKR capacity, and alleviating the information attenuation problem caused by decoherence in quantum networks [168,169,170].

Although there has been significant progress in the research of multi-user QKD networks [171,172,173,174,175,176] in recent years, bipartite QKD communication remains the main research subject. How to efficiently construct QKD networks with complex topological structures and effectively develop and utilize the quantum resources is currently an urgent problem to be solved. The study on network nonlocality characteristics helps to understand the impact of network topology on nonlocality and can construct more novel inequalities, providing new ideas and impetus for the development of QKD networks. Networks with different topologies have different roles in QKD networks. Star networks are used for access networks [129], mesh and cyclic networks are used for metropolitan area networks [134], and chain networks are used for long-distance backbone networks [134]. Triangle networks are most suitable for QKD models [177]. Tree networks can achieve hierarchical management of key resources [178]. The progress of the network nonlocality is summarized in the following aspects.

### 4.1. Generalized n-Locality Inequalities

Nonlocality is a process in which qubits maintain strong correlations even when separated by large distances and no signaling. When entangled qubits are used in quantum operations, the statistical correlations of the results will be inexplicable by classical means, which is named nonlocality. In the early stages, research on network nonlocality usually revolved around networks with a specific number of nodes and limited inputs. The nonlocality of a star network with five nodes and four independent sources has been verified in a bi-local scenario and beyond [179], which demonstrates that the device independence is one of the characteristics of network nonlocality, proving the scalability of star networks. However, there are many more nodes and sources in real-world networks, and the security and stability issues after network expansion have not yet been thoroughly solved. The proposal of generalized inequalities is beneficial for the research and development of quantum networks. The generalized *n*-locality inequalities for star, chain, tree-tensor, and cyclic topology are derived, respectively.

In [180], the number of the binary-outcome measurements in the edge nodes was extended from two times to arbitrary times, which unveiled an interesting phenomenon. When the number of measurements is larger than three, the non-*n*-locality in the star network may be activated by multi-copies of a two-qubit entangled state, which provides a general idea for stimulating or verifying network nonlocality. The generalization of *n*-locality inequalities in chain networks is reported with an arbitrary number of inputs to the edge nodes [181], which is quite similar to the process in star networks and shown in Equation (5).(5)$${\left(\beta_{n}\right)}_{n-local}\leq \sum_{l=0}^{\left\lfloor n/2\right\rfloor }\left(\begin{array}{c} n\\ l \end{array}\right)(n-2l)$$

Here, *n* is the dimension of the locality, and *l* is the binary-outcome observables per party. When *l* is greater than three, higher-dimensional entanglement systems are needed to ensure inequality violation maximization. During the analysis of the nonlocality in the chain network, a method named elegant sum of squares (SOS) [182] is adopted to derive the optimal quantum violation value of inequalities, which is independent of the dimensionality of the quantum system and suitable for customizing Bell inequalities for an arbitrary given quantum state [183].

A triangle network can be treated as the simplest cyclic network in terms of structure. The generalized cyclic network nonlocality with any number of nodes has been reported [184], with a new type of Bell inequality shown in Equation (6) to certify the genuine network nonlocality, which will be reviewed in a subsequent section of this paper and solves the nonlocality problem of even-cycle networks left in [185]. A device-independent model has been proposed [184] to break through the limitations of the assumption on independent sources, which is one of the major obstacles in quantum cryptography. The previous method required assuming that the quantum source was independent and trustworthy, while devices in the actual network may be maliciously connected. The proposed Bell inequality is very interesting, which is the sum of multipartite CHSH-type inequalities, and the inequality will return to the conventional CHSH inequality when *n* is 2. This new inequality solves the verification problem of complex network topology such as even-cycle networks and is a great extension to previous research [185,186,187].(6)$${\bar{\omega }}_{q}=\sum_{i=1}^{\left\lfloor \frac{n}{2}\right\rfloor }L_{i}\leq 2\sqrt[{n}]{2}-4\sqrt{2}+2\;\mathrm{with}\;\mathrm{the}\;\mathrm{largest}\;\mathrm{bound}\;\mathrm{of}\;2\sqrt[{n}]{2}-2\sqrt{2}$$

Here, ${\bar{\omega }}_{q}$ is defined as the quantum violation, *L_i_* denotes the summation of multipartite CHSH-type quantities, and *n* is the dimension of the locality.

Increasing the number of binary measurements in each node is the general method to generalize the Bell inequalities for tree-tensor networks [147] based on the analysis in [188]. The measurement times of edge nodes are greater than or equal to those of intermediate nodes, and all of them are larger than two times. By generalizing the CHSH inequality to the two-forked tree-tensor network, 6-local inequalities are constructed, and the inequality violations under different entangled states are analyzed. The nonlocality for the tree-tensor network with any number of forks has been further revealed [189] to be in the bipartite quantum states. The generalized model, shown in Equation (7), is quite flexible, and can transform into a two-forked tree, star, or chain network with specific configurations of the network parameters.(7)$${\left|I_{i_{1},i_{2},\ldots ,i_{t_{n-1}},0}\right|}^{\frac{1}{p_{n}}}+{\left|I_{j_{1},j_{2},\ldots ,j_{t_{n-1}},1}\right|}^{\frac{1}{p_{n}}}\leq 1$$

Here, *p_n_* is the number of end nodes, and the number of independent bipartite sources is *t_n_*_−1_.

The real network topology is often a fusion of two or more network structures, such as star networks with chain-like branches [190]. The parameters for determining the main structure of a quantum network include the number of independent sources (denoted by S), the number of entangled particles owned by each intermediate node (denoted by P), and the number of edge nodes (denoted by E), which corresponds to a cyclic network when E equals 0. The generalized *n*-local inequality in the (S, P, E)-type quantum network, which can cover multiple network topologies and for the first time achieves a decentralized and asymmetric universal network architecture, has been obtained [191] and is shown below. The unified description method of network topology can reduce ambiguity, making the expansion and analysis of quantum networks more intuitive and flexible. The architecture model does not emphasize the role of the central nodes, avoiding dependence on the central nodes. On this basis, it is found that the nonlocality of quantum networks is only determined by the degree of entanglement of the sources and is independent of the choice of measurement operation or the type of measurement operator, which is shown in Equation (8).(8)$${\left|I_{X}^{0}\right|}^{\frac{1}{p}}+{\left|I_{X^{’}}^{1}\right|}^{\frac{1}{p}}\leq 1$$

Here, X and X′ are the sets of inputs for the intermediate nodes, and *p* is the number of the end nodes.

The Bell inequalities above are nonlinear, which can achieve the Tsirelson bound of |B*_n_*| ≤ *n*! in a non-*n*-locality scenario [192,193]. Thus, the nonlinear Bell inequalities can describe the properties of entangled states more accurately compared to linear Bell inequalities and are especially suitable for noisy quantum communication scenarios, which is demonstrated by experiment in [194].

### 4.2. Sharing Network Nonlocality

The sharing of network nonlocality is a process of distributing and utilizing nonlocality resources, such as entangled states, in quantum networks to support applications, such as quantum communication and distributed computing. The sharing process typically relies on network topology design, such as (*n*, *m*, *p*)-type frameworks [191], and protocol optimization, such as entanglement swapping [151,195,196,197,198,199,200,201,202,203], which requires the coordination of multi-node measurement strategies, such as Pauli measurements [204,205], to maintain nonlocality during the process. Nonlocality sharing in star [164,190], chain [206], tree-tensor [207], and triangle networks [208,209] has been reported, respectively. The number of users in star networks can be increased through chain-connected edge nodes, which is the fusion of chain and star networks, and the network nonlocality is shared via weak measurements in the intermediate nodes and strong measurements in the last node [190]. The Munshi–Kumar–Pan (MKP) inequality in [180] and shown in Equation (9) is suggested to be used to reveal the network nonlocality sharing when the number of inputs is greater than three.(9)$${\sum_{i=1}^{2^{m-1}}\left|\prod_{k=1}^{n}\left[\sum_{x_{k}=1}^{m}{\left(-1\right)}^{y_{x_{k}}^{i}}A_{x_{k}}^{k}\right]B_{i}\right|}^{\frac{1}{n}}\leq \sum_{j=0}^{\left\lfloor \frac{m}{2}\right\rfloor }\left(\begin{array}{c} m\\ j \end{array}\right)\left(m-2j\right),\left(\begin{array}{c} m\\ j \end{array}\right)=\frac{m!}{j!\left(m-j\right)!}$$

Here, *n* is the number of edge observers and the number of independent sources, *m* is the number of dichotomic measurements performed by every observer. $A_{x_{k}}^{k}$ is the binary-outcome of the *x_k_*-th measurement performed by the *k*-th observer ($\forall x_{k}\in \left[m\right]$, for any *k*, and $k\in \left[m\right]$). Projective measurement has been proposed in [210,211] and used in [151] to replace the weak measurement to share and maintain the nonlocality in the network. The projective measurement is one of the strong measurements, which distributes nonlocality through the network by using classical randomness sharing for any high-dimensional pure state without breaking the entangled states and avoiding additional loopholes [103,104] by weak measurements.

In the research of quantum nonlocality sharing, protocol optimization significantly improves the sharing efficiency and security of nonlocality associations by improving entanglement mechanisms [196,197,198,199,202,203,206,207], measurement strategies [151,206,207,208,209,210,211], resource allocation [200], and network architecture design [201].

The entanglement protocol optimized for specific entangled states [198] and specific network architectures [202,206,207] can improve the sharing efficiency and save network resources. In response to the problems of weak measurement faced by the W state, an entangled state purification method is proposed [198], which provides an improvement idea for other weak measurement schemes [200].

The all-optical entanglement-swapping scheme can avoid photoelectric conversion [196,198] and simplify the swapping protocol and the structure of quantum relay [196]. Relevant experiments [203] have achieved entanglement swapping based on time-bin encoded photons with an average fidelity of 87%, which can support QKD applications. The all-optical scheme is beneficial for further expanding the communication bandwidth of quantum networks and has the potential to be directly extended to mixed entanglement and many-body entanglement swapping [196,199]. The fidelity, denoted by *F*, of many-body entanglement swapping can be calculated using Equation (10).(10)$$F=\left\langle \Psi_{T}|\Psi_{AB}\right\rangle =\frac{1}{\sqrt{P_{E}}}\sum_{k}\lambda_{k}^{3/2}r_{kk}^{E},and\sum_{ij}{\left|r_{ij}^{E}\right|}^{2}\leq 1$$

Here, *P*_E_ is the probability of outcome for Eve, who is denoted by the superscripts and subscripts of E, and *λ_k_* > 0 are the Schmidt coefficients.

The realization of clock synchronization in all-optical entanglement schemes is quite difficult [196,197,198]. A tree structure can be used in the clock distribution network, combining a central controller with a layered network structure, to achieve time synchronization [202]. And the hierarchical distributed network is the foundation for optimizing the quantum memory allocation and entanglement resource allocation at the link level [200,201].

Many-body entanglement is currently a hot topic in quantum research [212,213,214,215,216]. In order to overcome the limitations of two-body entanglement swapping [217], the partitioned general many-body states [199] and the sequential weak measurement method [208,209] based on positive operator valued measurement (POVM) [218,219] have been proposed. The protocol cost of [199] is determined by the third Rényi entanglement entropy [220,221] of the partitioning, which is independent of system scale and supports fault-tolerant entanglement swapping [222,223,224], making it suitable for large-scale QKD networks. The sequential weak measurement method, by violating the Mermin inequality shown in Equation (11) [208], can infinitely share nonlocality [209], which means a single Alice and Bob can share nonlocality with any number of independent Charlies, breaking through the upper bound of six Charlies in [208]. In addition, in the process of nonlocal sharing, the use of projection measurement instead of weak measurement has been verified [151,210,211], which can achieve Bell nonlocality sharing of any high-dimensional binary pure state [211].(11)$$S_{m}=\left|\bar{C_{000}^{m}}+\bar{C_{100}^{m}}-\bar{C_{010}^{m}}+\bar{C_{110}^{m}}+\bar{C_{001}^{m}}-\bar{C_{101}^{m}}+\bar{C_{011}^{m}}+\bar{C_{111}^{m}}\right|$$

Here, Equation (11) is the Svetlichny inequality for Alice, Bob, and the *m*-th Charlie.

The nonlocality sharing achieved through classical randomness sharing (probability mixing of different projection measurement strategies) extends the research framework on two-qubit states [210]. Another nonlocality called genuine non-signal nonlocality, which can be shared an arbitrary number of times by the violation of the non-signal inequality shown in Equation (12), has been detected [209]. The proof of the nonlocality sharing of high-dimensional systems by adjusting the measurement operator inspires the serialization implementation of other quantum protocols based on projection measurement, such as quantum steering [225,226,227] and contextuality [228,229,230]. But the sharing of Svetlichny nonlocality [231,232,233,234,235] in a setting of more than two Charlies with a single Alice and Bob is still a problem not yet solved.(12)$$\langle X_{0}Z_{0}\rangle +\langle Y_{0}Z_{0}\rangle +\langle X_{1}Y_{0}\rangle -\langle X_{0}{Y_{1}Z}_{1}\rangle +\langle X_{1}{Y_{1}Z}_{1}\rangle \leq 3\;and\langle X_{i}Y_{j}\rangle =\sum_{AB}{\left(-1\right)}^{A+B}P\left(AB|X_{i}Y_{j}\right)$$

### 4.3. Detecting/Verifying Network Nonlocality

A quantum network must hold a degree of nonlocality to ensure its ability to perform quantum functions, such as sharing [151,164,190,206,207,208,209,211] or recycling nonlocality [210,236]. The detection and verification of network nonlocality aim to verify whether there are non-classical correlations in the network, usually by violating the *n*-local inequalities to prove that network behavior cannot be explained by LHV theory. Based on mathematical inequalities and experimental verification, the strength of nonlocality can be quantified by calculating the degree of quantum violation, which ensures that the quantum communications are free from classical eavesdropping. The detection and verification of network nonlocality in different network topologies have been reported, such as triangle [177,237,238,239,240,241,242], star [163,180,243,244], multi-star [245], chain [163,246,247], and tree [153,248].

A set of nonlinear Bell-type inequalities has been proposed for triangle networks [237] to detect non-tri-local correlations shown in Equation (13) by using Bloch matrix representation [249,250] of the two-qubit state, which is generalized to a cyclic network with n edges and n vertices.(13)$$\begin{array}{l} \sqrt{\left|2\alpha_{1}^{2}-\alpha_{2}^{2}\right|}+\sqrt{\left|\alpha_{1}^{6}-\alpha_{1}^{4}\alpha_{2}^{2}+\alpha_{2}^{6}+\alpha_{1}^{2}\left(3-\alpha_{2}^{4}\right)\right|}\leq 2^{3/2}\;\mathrm{for}\;\mathrm{triangle}\;\mathrm{network}\\ \sqrt{\left|p_{2}\left[p_{2}-3p_{2}\alpha_{2}^{2}-\left(1-p_{2}\right)\alpha_{2}^{4}\right]\right|}+\sqrt{\left|p_{2}^{2}\left(p_{2}-\alpha_{2}^{2}+4\alpha_{2}^{4}\right)\right|}>2^{3/2}\;\mathrm{for}\;\mathrm{violation}\;\mathrm{with}\;\mathrm{state}\;\mathrm{noise}\\ \left|p_{3}^{2}\left(\alpha_{2}^{2}-4\alpha_{2}^{4}+p_{3}\right)\right|+\left|p_{3}\left[p_{3}-3\alpha_{2}^{2}p_{3}+\alpha_{2}^{4}\left(1+p_{3}\right)\right]\right|>2^{3/2}\;\mathrm{for}\;\mathrm{violation}\;\mathrm{with}\;\mathrm{channel}\;\mathrm{noise}\\ \sqrt{\left|p_{4}^{2}\left[3p_{4}\alpha_{2}^{2}-p_{4}-\left(1-p_{4}\right)\alpha_{2}^{4}\right]\right|}+\sqrt{p_{4}^{3}\left|\left(1-\alpha_{2}^{2}+4\alpha_{2}^{4}\right)\right|}>2^{3/2}\;\mathrm{for}\;\mathrm{violation}\;\mathrm{with}\;\mathrm{measurement}\;\mathrm{noise} \end{array}$$

Here, *α*_1_ and *α*_2_ are the measurement parameters with 1 > *α*_1_, *α*_2_ > 0, and *α*_1_^2^ + *α*_2_^2^ = 1. And *p*_2_, *p*_3,_ and *p*_4_ are noise parameters for state, channel, and measurement, respectively.

The product states do not support the detection of non-tri-locality in a network; thus, the pure state detection scheme is suggested. And the non-tri-locality in the *n*-local star network is analyzed [244] by involving stochastic local operations assisted with classic communications. In the sequential *n*-local star network, the upper bound of the *n*-local inequality is given in Equation (14).(14)$$B_{n-star}^{\left(sequential\right)}=2\sqrt{\prod_{i=1}^{n}{\left(W_{i1}^{\left(f\right)}\right)}^{2/n}+\prod_{i=1}^{n}{\left(W_{i2}^{\left(f\right)}\right)}^{2/n}}and\;\forall i,W_{i1}^{\left(f\right)}\geq W_{i2}^{\left(f\right)}$$

Here, $W_{i1}^{\left(f\right)}$ and $W_{i2}^{\left(f\right)}$ denote the ordered singular values of the correlation tensor of the state.

A nonlocality certification method for triangle networks based on elegant joint measurement (EJM) [251,252], which is also nonlocal [242], has been proposed [177] to reveal the correlations between the symmetry of the measurement basis and quantum nonlocality. A generalized EJM basis has been constructed to achieve continuous control from partially entangled states to maximally entangled states. Non-tri-locality has been achieved in triangle networks with only partial entanglement states [253,254] for the first time. The probability criteria for non-*n*-locality have been derived by generalization to a closed polygon network [177,237] shown in Equation (15), in which *n* ≥ 4. The nonlocality still exists within the noise threshold (V ∈ (0.86,1)), by comprehensive analysis of the Werner state and mixed state [177], which is consistent with the theoretical analysis of [237].(15)$${\left|I_{1,n}\right|}^{\frac{1}{2}}+{\left|I_{2,n}\right|}^{\frac{1}{2}}\leq 1$$

In order to optimize the allocation of network resources and explore the quantum theory, the nonlocality minimization of triangle networks [238,239,240], which focuses on the scenario of triangle networks without inputs, has been studied in depth. Inflation [24] and a neural network (an LHV net) have been used to verify the minimal quantum nonlocality [238], where inflation is used to generate Bell inequalities and the LHV net is used to evaluate noise robustness. The coarse graining method [255,256,257,258] has been used to simplify the output cardinality, which is adaptive to LHV-net and inflation. The critical visibilities of different output cardinalities (3-3-3 and 3-3-2) have been derived. The variational Lovasz local lemma [259,260] have been used to analyze the 2-2-2 triangle. The local model boundaries of symmetric distribution in the minimal triangle network have been characterized thoroughly [240] by using Gröbner basis [261,262] and hybrid calculations, which can be used to solve the problem of variable elimination in high-dimensional parameter space, and different types of boundary models, as well as the corresponding Bell inequalities, have been proposed. The cardinality of hidden variables can be reduced to three, instead of six as in [263], which significantly reduces the complexity of LHV models.

The nonlocality without inputs in the minimal triangle network has been realized experimentally for the first time in [239] with the arbitrary small level of independence between sources, which supports the viewpoints proposed in [264], and the sketch of the experimental setup is shown in Figure 5. The experiment was carried out by using a single source of broadband multiplexed entangled-photon states [265,266,267,268,269] equivalent to three independent entanglement sources, which proves that even with strongly correlated classical sources, quantum nonlocality can still be observed.

The correlator space of the symmetric, network-local distributions generated in the minimal triangle scenario is constructed and depicted in [240], which represents the boundaries of multiple inequalities using multiple irregular faces within a triangular pyramid. Several different distributions are presented in visual form in [240], where, *GHZ* is a shared random bit distribution, *W* is the uniform mixture distribution, $\bar{W}$ is the inverse of *W*, *U* is the uniform random distribution, and *D*_+_ and *D*_−_ are two deterministic distributions, all of which are related to the three symmetrized correlators defined in Equation (16).(16)$$E_{1}=\sum_{a,b,c\text{∈}\left\{-1,1\right\}}ap\left(a,b,c\right),\;E_{2}=\sum_{a,b,c\text{∈}\left\{-1,1\right\}}abp\left(a,b,c\right),\;E_{3}=\sum_{a,b,c\text{∈}\left\{-1,1\right\}}abcp\left(a,b,c\right)$$

Here, *p*(*a*, *b*, *c*) is the joint probability distribution of the binary output of *a*, *b*, and *c*.

To close the loopholes in detection, the high-dimensional photonics state of 2002 is used instead of a single photon to enhance the tolerance of the system to the transmission loss [241], which certifies noise robustness up to 10.3% single photon loss and full photon loss noise of 0.14%. The new framework is based on a complementary approach of linear programming and neural network (LHV Net) [238,241] to validate the results, and its noise model analysis ability is superior to the methods of token-counting and noise parity token-counting [238,270,271]. But further analysis of the singular point found at η = 0.6 is needed.

Multi-star-shaped topology is commonly used in QKD networks [129,134]. The Bell-type inequalities for multi-star networks have been proposed [272,273,274] and generalized in strong locality assumptions [245], as shown in Equation (17).(17)$$\sum_{s=1}^{2^{n-1}}{\left|I_{i_{1}^{s},\ldots ,i_{2m}^{s},s}\right|}^{\frac{1}{m^{2}}}\leq \sum_{p=0}^{\left\lfloor n/2\right\rfloor }\left(\begin{array}{c} n\\ p \end{array}\right)\left(n-2p\right)$$

Here, *m* is the number of nodes in each layer of the star network and *n* is the number of measurements in each node.

The optimal quantum violation can be achieved by multi-copies of entanglement when the measurement exceeds three times, which is consistent with [180]. If a multi-star network is divided by hierarchy, treating the core node as the root node, then the multi-star network is transformed into a tree network, which is used to verify the nonlocality of the *k*-forked tree network (*k* > 1), and the inequalities for a quantum network with multiple parties are derived [153]. The *n*-local inequalities are generalized from bi-locality to multiple sources with different inputs, and the optimal quantum violations of star and chain networks are deduced using the SOS method [180,247]. A powerful tool of deep learning has been introduced to detect the nonlocality in the chain network in the Werner state [275] with the non-*n*-locality quantifier, which can collect data across different network scenarios [246], which can be generalized to other types of networks.

Quantum networks are inevitably affected by noise, including entanglement generation noise, measurement error noise, and communication noise, which can be divided into amplitude-damping noise and phase-damping noise [248,276]. The effects of different noises are independent and additive, and the impact of measurement noise is the greatest, followed by amplitude-damping noise. Entanglement generation noise is relatively small, and especially the phase-damping noise does not disrupt the nonlocal correlations solely [243,248]. The dynamics of communication noise can be analyzed, and the process of the fidelity of the Bell states changing over time under the amplitude-damping and phase-damping noise can be calculated using Equation (18) [276].(18)$$F_{\left.|\phi^{\pm }\right\rangle }=\frac{1}{4}\left(2+2e^{-t\left(\tau_{1}+\tau_{2}\right)}+2e^{-\frac{t}{2}\left(\tau_{1}+\tau_{2}+4\gamma_{1}+4\gamma_{2}\right)}-e^{-t\tau_{1}}-e^{-t\tau_{2}}\right)\;F_{\left.|\Psi^{\pm }\right\rangle }=\frac{1}{4}\left(2e^{-\frac{t}{2}\left(\tau_{1}+\tau_{2}+4\gamma_{1}+4\gamma_{2}\right)}+e^{-t\tau_{1}}+e^{-t\tau_{2}}\right)\;\left.|\phi^{\pm }\right\rangle =\frac{1}{\sqrt{2}}\left(\left.|00\right\rangle \pm \left.|11\right\rangle \right),\left.|\Psi^{\pm }\right\rangle =\frac{1}{\sqrt{2}}\left(\left.|01\right\rangle \pm \left.|10\right\rangle \right)$$

Here, *τ* and *γ* are the attenuation rates of amplitude-damping noise and phase-damping noise, respectively.

The entanglement generation noise is introduced by the imperfection of the quantum gates [248], which will degenerate the purity of the entanglement states, and its influence on the Werner state [177,181,237] and GHZ state [277] has been reported. The noise parameter threshold for quantum violations in the general domain has increased from 0.707 for bi-locality to 0.908 for tri-locality [181], which is quite similar to 0.892 in [237], which means more measurements are needed to support high-dimension entanglement. The measurement method will affect the degree of quantum violations; for example, the degree of violation of CHSH inequality caused by projection measurement (approximately 2.108) is lower than that of non-sharp measurement schemes (approximately 2.263) [210]. Meanwhile, the selection of measurement basis vectors is also crucial. Both Werner states and partially entangled states may achieve non-tri-locality under specific measurement basis vectors, while other basis vectors may reduce or lose the nonlocality [177]. The GHZ states are sensitive to noise, while the robustness of high-dimensional entangled cluster states is higher [277,278]. The continuous variable cluster state quantum entanglement based on integrated optical quantum chips has been achieved recently for the first time [279], and the experiment scheme is shown in Figure 6.

The robustness of quantum networks to noise depends on the topology, size, and entanglement distribution within the network [278]. The noise accumulation in cyclic networks is the most severe, which will decline the nonlocality of the cyclic network rapidly and disrupt the correlations between nodes. The ability of forked tree networks to maintain entanglement in noisy environments is determined by the fork number of a determinate forked tree network or the number of independent sources in the last layer of the indeterminate forked tree networks [248]. When the fork number is larger than 14, the maintenance of entanglement is independent of the number of sources, which means the number of sources can be increased to infinite [248]. While in the indeterminate forked tree networks, the more sources in the last layer, the stronger the persistence of entanglement [248]. In a white noise environment, a better noise level is needed to ensure quantum violation when the input of edge nodes of a chain network reaches three or more, which means more noise is introduced [181]. The star network has the strongest ability to resist consistent noise [243]. The non-ideal characteristics of sources will introduce extra phase noise, which will usually become more serious as the number of sources increases and affect the violation of nonlocal inequalities.

However, the nonlocal inequality of star networks is independent of the number of sources, which is theoretically immune to the noise introduced by sources [243]. Thus, the star networks gain higher priority in the construction of quantum networks. The network topology will change with the number and distribution of sources. If the network topology is unknown, how do you determine the topology and verify the existence of the network nonlocality? The answer is the testing of Bell inequalities. The topological stability of network nonlocality is demonstrated [280] by the rigidity of token counting [270] and the neural network oracle [281] even if only a partial network is known or trusted. By constructing more precise Bell-type inequalities and quantifying the minimum number of quantum sources in a network, the distribution of quantum sources and classical sources can be detected in quantum networks with the help of hierarchical nonlocality of the quantum networks [282]. The improved Bell inequalities shown in Equation (19) provide more accurate upper bounds of the inequality [152], achieving precise counting of the number of quantum sources. At the same time, the *l*-level quantum network locality is generalized to a network with an arbitrary topology and an arbitrary number of parties [282] on the basis of [283].(19)$${\left|I\right|}^{\frac{1}{n}}+{\left|J\right|}^{\frac{1}{n}}\leq 2^{\frac{n-l}{2n}},\;when\;l>\frac{n}{2}\;I=\langle B_{0}\prod_{i=1}^{n}A_{x_{i}}^{+}\rangle \;and\;J=\langle B_{1}\prod_{i=1}^{n}A_{x_{i}}^{-}\rangle \;A_{x_{i}}^{\pm }=\frac{A_{x_{i}=0}\pm A_{x_{i}=1}}{2}$$

Here, *l* is the number of sources and *n* is the number of branches in a star network. A*_xi_* represents the input of the *i*-th branch, while the outcome of B, the central node of a star network, is represented by the subscripts 0 and 1.

Improving the quantum violation degree and accuracy of quantum networks can effectively enhance the noise tolerance and application scenarios, such as constructing new or modified quantum inequalities [163,166,284,285,286], improving testing schemes [45,287], and optimizing entanglement source performance [93,239,288,289].

### 4.4. Full Network Nonlocality

Bell nonlocality in networks brings novel forms of entanglement to the forefront of the physical process, which means the analysis of network nonlocality should not fall back on the long-established ideas and tools for standard Bell inequality experiments. Thus, the research on network nonlocality has turned out to be a formidable challenge that demands new lines of thought. The violation of traditional Bell’s inequality in networks, such as the bi-local inequality, only proves the existence of at least one non-classical source in the network. While a quantum network usually consists of multiple independent sources that can generate entangled particles and distribute them to a set of parties, which is quite different from traditional Bell experiments with only one source and can generate new quantum correlations. Before 2022, the representation of quantum network nonlocality could easily revert back to the standard Bell’s nonlocality expression [290], making the role of the network in quantum communication trivial, which is of course not the truth. To solve this problem, a stricter definition, named full network nonlocality (FNN), has been proposed [24]. FNN requires that all sources must be non-classical, which means that all sources in the network cannot be simulated by classical variables, even if other sources are no-signaling-limited, which is stricter than standard network nonlocality because it excludes the possibility of any part of the network using classical sources and provides a new perspective to understand the nonlocality in quantum networks. It is similar to adding the harsh condition of fully connected [291] on the basis of traditional Bell nonlocality. The most interesting thing is that they discovered the famous Branciard inequality [292] shown in Equation (20), which is applied to star and chain networks with multiple sources, can be falsified by a single nonlocal source, indicating that there are loopholes within the existing testing tools and highlighting the necessity of defining and studying FNN [24].(20)$$\sqrt{I_{AC}}+\sqrt{I_{BC}}\leq 1$$

Here, *I*_AC_ is the correlation between Alice and Charlie, and *I*_BC_ is between Bob and Charlie.

To solve this loophole, a tight inequality is proved in a star network with three sources, as shown in Equation (21) [24].(21)$$\left|I_{1}\right|+\left|I_{2}\right|+\left|I_{3}\right|+\left|I_{4}\right|\leq 1$$

Here, *I_i_* (*i* = 1,2,3,4) is the correlation for the star network with three sources.

A star network with three branches has been constructed to analyze the characteristics of FNN with new inequalities (called KGT inequalities [24,293,294], shown in Equation (22)) and the inflation method [295], which is used in causal inference with hidden local variables and has advanced capability of witnessing incompatibility compared to other causal inference schemes [296,297]. When each source in the three-branch star network emits a Werner state, the FNN can be achieved as long as the visibility exceeds 89.1% with the noise tolerance of about 10.9% [24].(22)$$R_{C-NS}=2\langle A_{0}B_{1}C_{0}\rangle -2\langle A_{0}B_{1}C_{1}\rangle +2\langle A_{1}B_{0}C_{0}\rangle +\langle A_{1}B_{0}C_{1}\rangle -\langle B_{0}\rangle \;+\langle C_{1}\rangle \left|\langle A_{1}B_{0}\rangle +\langle B_{0}C_{0}\rangle -\langle C_{0}\rangle \right|\leq 3\;R_{NS-C}=2\langle A_{0}B_{1}C_{0}\rangle -2\langle A_{0}B_{1}C_{1}\rangle +\langle A_{1}B_{0}C_{0}\rangle +2\langle A_{1}B_{0}C_{1}\rangle -\langle B_{0}\rangle +\langle A_{1}\rangle \langle A_{1}B_{0}\rangle \;+\langle A_{1}\rangle \langle B_{0}C_{1}\rangle +\langle A_{1}\rangle \langle C_{0}\rangle -\langle A_{1}\rangle \langle C_{1}\rangle -\langle A_{1}\rangle \langle A_{1}\rangle \leq 3$$

With the help of violations of KGT inequalities, the FNN is experimentally demonstrated in a bi-local scenario [293,298], both of which are based on the scheme of partial BSM in the central node and single photon polarization analysis in the edge nodes. By optimizing the test condition and process, multiple loopholes are closed and greater quantum violations (3.321 and 3.356 for R_C-NS_ and R_NS-C_, respectively) are achieved [298]. Then a six-photon experiment is carried out to certify the FNN in a three-branch star network [299], which is more complex than bi-local networks and requires stronger non-classical certification, and the experimental setup is shown in Figure 7.

The central node performs GHZ projection measurement, and the branch nodes perform random basis measurements. FNN is successfully verified with the visibility of 0.882 [299]. The inflation method proposed in [24] is implemented in a more complex form in experiments, and the corresponding inequalities are derived, which is a nontrivial generalization of CHSH inequality [299]. In a quantum network, inflation is a technique by which the sources and measurement devices of a network are copied several times and arranged in different configurations, which introduces constraints that do not appear directly in the original network, especially the independence or cloning relationship between different replicas. To some extent, inflation on the network is quite similar to adding a garbage state in the quantum algorithm [300]. Other quantum technologies can also be used to verify FNN, such as the experimental implementation of KGT inequality violation based on hyper entanglement [301,302] and EJM [303].

In order to solve the difficulty of FNN verification in complex networks, the complex network can be divided into subnets, and nonlinear inequalities corresponding to each subnet can be designed. The violations of KGT inequalities can be achieved by measuring the maximum entangled states and generalized Bell state to verify FNN [156]. For example, in a four-party chain network, FNN is validated through adjacent tripartite subnetworks. The verification of FNN in tree networks is realized by the hierarchical certification method, which gradually transits from standard network nonlocality to FNN by distinguishing the number of classical sources in the network [294]. In this process, in order to adapt to different network scenarios, nonlinear inequalities are generalized hierarchically, and linear inequalities are generalized to arbitrary lengths of chain networks. The Born rules [304,305] can be used to reduce the inequality violation thresholds of FNN certification. Thus, the FNN of a tree network is generalized to *l*-level quantum network nonlocality [152], which means that at least *l* sources are required in the network to distribute classical physical systems, and FNN corresponds to the special case of *l* = 1. The *l*-level quantum network locality inequalities for *k*-forked tree networks, acyclic networks, cyclic networks, and general networks are generalized [153] based on the hierarchical scheme proposed in [152] and the method of maximal independent-node number, which is an NP-hard problem [306,307]. A Bell-type witness operator, which is hybridized by the Mermin operator [208,308] and the CHSH operator [309], is proposed and shown in Equation (23) to unifiedly certify the topology and nonlocality of the triangle network [146], which can identify five different topologies of triangle networks, including the three-particle GHZ state. In the experiments of different photonic triangle networks, the measurement results of different topologies significantly exceeded the classical limits, such as the average measurement value of the witness operator in a triangle network with three sources of entangled pairs, which can reach 25.2077 ± 0.2106, while the theoretical limit is $18\sqrt{2}\approx 25.4558$ [146].(23)$$\begin{array}{cc} B≔2[A_{0}B_{0}C_{0} & +A_{0}B_{0}C_{1}+A_{1}B_{1}C_{0}-A_{1}B_{1}C_{1}+A_{2}B_{2}C_{2}+A_{2}B_{3}C_{2}+A_{3}B_{2}C_{3}\\ & -A_{3}B_{3}C_{3}+A_{4}B_{4}C_{4}+A_{5}B_{4}C_{4}+A_{4}B_{5}C_{5}-A_{5}B_{5}C_{5}]+A_{0}C_{0}\\ & +A_{0}C_{1}+A_{1}C_{0}-A_{1}C_{1}+B_{2}C_{2}+B_{3}C_{2}+B_{2}C_{3}-B_{3}C_{3}+A_{4}B_{4}\\ & +A_{5}B_{4}+A_{4}B_{5}-A_{5}B_{5} \end{array}$$

Different sharing strategies of FNN are proposed and discussed in [294], including passive and active sharing. The passive FNN sharing is impossible to achieve, while the active sharing can be achieved, but it is sensitive to noise and requires deep collaboration among intermediate observers. During the analysis, the maximum violation of KGT inequalities in the bi-local scenario can reach 3.6055 in theory, which is better than the initial value of 3.5355 [24].

Research on measurement dependence is a scientific method that reveals objective laws by quantifying covariate relationships between variables and plays a very important role in the study of network nonlocality [310,311,312]. In [312], it is found that the relaxation of the measurement selection independence of only one end party can classically simulate the maximum quantum violation. In a star network with four parties and three sources, it is only necessary to apply 92% measurement dependence to one of the end parties to reproduce the maximum quantum violation of FNN. The middle party can manipulate the entire network correlations by controlling the measurement dependence or randomness of an end party [312], providing a new perspective for understanding security loopholes in network communications. The optimization of the measurement scheme for FNN is proposed, and new inequalities for star and chain networks are constructed and generalized [313]. The new scheme does not require EJM [303], but only two output measurements, significantly improving the experimental friendliness. The quantum optimal violations are derived through analytical methods without assuming quantum system dimensions, and the device-independent self-testing [314,315,316] is supported during the process [313].

### 4.5. Genuine Network Nonlocality and Device-Independent QKD

The rigorous definition of genuine network nonlocality (GNN) [317], which has been mentioned in previous papers [185], is proposed to distinguish unique nonlocal correlations in quantum networks, especially in the multipartite entanglement scenarios, which are inherent to the network rather than a simple expansion of standard Bell nonlocality. GNN relies entirely on the network topology, which means the corresponding correlation cannot be explained by the combination of decomposed smaller nonlocal resources and globally shared random variables, and it is proved by self-testing that the network correlation can exist independently of Bell nonlocality [317,318]. GNN is experimentally certified in a multipartite quantum network that is subject to local operations and shared randomness (LOSR) by the experiment of inequality violations by the four-photon GHZ state in an inflated network [319].

The proposal of GNN provides a more reliable security foundation for QKD, especially in multi-party quantum networks [320,321]. Self-testing proposed in 2004 [322] is an important tool for validating GNN [157,317,323] and is the core method for verifying quantum devices within a black box framework [324,325], which relies solely on the input-output statistical correlation of the device to uniquely characterize internal quantum states and measurement operations, without the need for any prior calibration or internal information [320]. Self-testing provides theoretical support for cryptographic protocols in black box scenarios, ensuring security without relying on device trustworthiness, such as DI-QKD [326]. The degree of key security is proportional to that of Bell inequality violation, which directly relates Bell inequality to the security of QKD systems. Thus, the DI-QKD protocol has been proposed [327,328,329,330,331] and demonstrated [126,332], which directly uses the Bell inequality violation value to generate a key and provides security against the system defects during implementation. Thus, the research on GNN and self-testing provides a solid theoretical and experimental foundation for DI-QKD [157,310,323,333,334,335,336,337,338].

Improving the robustness of self-testing and studying security adversaries [339,340] in untrusted scenarios, such as broadcast scenarios, are the key to implementing DI-QKD. In broadcasting scenarios, untrusted receivers may eavesdrop on key information; therefore, more comprehensive measurement methods and more accurate discrimination bounds [341,342,343] are required. A protocol to certify the genuine multipartite correlations in the GHZ state in a network with dishonest parties has been proposed, and the family of *N*-partite Svetlichny inequalities is shown in Equation (24) [336,344,345].(24)$$\left|S_{N}^{\pm }\right|\leq 2^{N-1}\;in\;classical\;bound\;and\;\left|S_{N}^{\pm }\right|\leq 2^{N-1}\sqrt{2}\;in\;quantum\;bound$$

And if S_N_ and S_k_ are the values of the *N*-partite and *k*-partite Svetlichny inequality, respectively, which are achieved by the same strategy, these two values should satisfy the following inequality shown in Equation (25).(25)$$s_{k}\geq \frac{s_{N}}{2^{N-k}}$$

A lower bound on the extractability of the *k*-partite Svetlichny inequality is derived as Equation (26).(26)$$F_{DI}\left(s_{k}\right)\geq f_{k}s_{k}-\mu_{k}$$

Then, the fidelity should be certified by an honest party with Equation (27).(27)$$F_{DI}^{\left|D\right|=N-k-1}\geq f_{k}\frac{s_{N}}{2^{N-k}}-\mu_{k}$$

Here, *f_k_* and *µ_k_* are the coefficients that bound the extractability for the *k*-party Svetlichny inequalities in the standard Bell scenario. The impact of noise on inequalities is represented by *µ_k_*. The STOPI (self-testing from operator inequalities) method [343,346,347] is used to derive the device-independent lower bounds for fidelity, which ensures the reliability of the certification even if some sources are maliciously controlled. Further, a truly device-independent certification method is proposed [335], which relies solely on local operations by generalizing the Hardy-type nonlocality arguments [342,348,349] without the help of a network or measurements. The bound of the maximum success probability of Hardy’s argument for the *N*-partite system is shown in Equation (28).(28)$$p_{max}=\frac{t^{N}{\left(1-t\right)}^{N}}{1-t^{N}}$$

Here, *t* is the positive root of x*^N^*^+1^ − 2x + 1 except for 1. The maximum success probability of the tripartite Hardy argument in a device-independent scenario is 0.018 without noise, and the lower bound of 1-*ε*^2/3^, with noise parameter *ε* has been derived [335]. The tight upper bound of genuine nonlocality of four-partite has also been derived [234] using the Seevinck–Svetlichny inequality [345,350], which belongs to the Svetlichny-type inequality. The four-particle Seevinck–Svetlichny operator is defined as Equation (29).(29)$$\begin{array}{cc} {SS}_{4} & =\left[A⊗B-A^{\prime }⊗B^{\prime }\right]⊗\left[\left(C-C^{\prime }\right)⊗D-\left(C+C^{\prime }\right)⊗D^{\prime }\right]\\ & -\left[A^{\prime }⊗B+A⊗B^{\prime }\right]⊗\left[\left(C+C^{\prime }\right)⊗D-\left(C-C^{\prime }\right)⊗D^{\prime }\right] \end{array}$$

Here, the observables are *X* = *A*, *A*′; *B*, *B*′; *C*, *C*′; *D*, *D*′. And the *SS*_4_ is bounded by the inequality shown in Equation (30).(30)$$V\left({SS}_{4}\right)=max\left|{\langle {SS}_{4}\rangle }_{\rho }\right|\leq 4\sqrt{2}\lambda_{max},{\langle {SS}_{4}\rangle }_{\rho }=Tr\left({SS}_{4}\rho \right)$$

Here, *λ*_max_ is the largest singular value of the correlation matrix and ρ represents any *N*-qubit state.

The bound for genuine nonlocality before local filtering is 0.707 and can be optimized to 0.201 after filtering, both of which are derived in the presence of noise. And the hypothesis of the Seevinck–Svetlichny operator bound for arbitrary *N*-particle state is shown in Equation (31), which is proved by the generalized Svetlichny operator [351].(31)$$V\left({SS}_{N}\right)=max\left|{\langle {SS}_{N}\rangle }_{\rho }\right|\leq {\left(\sqrt{2}\right)}^{N+1}\lambda_{max}$$

The entanglement-free nonlocality provides a physical basis for device-independent solutions that do not require entanglement resources while avoiding the dependence on the model of the quantum devices. In [317], genuine network quantum nonlocality without entanglement is speculated to exist and is demonstrated in subsequent research [323,352,353,354]. By constructing different types of genuine hidden nonlocality and activating them with the LOCC (local operations and classical communication) protocol, the existence of entanglement-free genuine nonlocality is verified under specific conditions [355], such as pure states and specific dimensions, which solves the issues raised in [317,356]. On the other hand, measurements of genuine nonlocality can also be entanglement-free [357], which means that self-testing without entanglement in the network can generate genuine network quantum nonlocality using product states [323], and the effect of nonlocality without entanglement can be combined with quantum nonlocality theory in a device-independent framework [157,317].

Closing system loopholes is an important step in enhancing the security of DI-QKD systems [358,359]. The certification of genuine quantum nonlocality often relies on the pure states, which cannot be met in practical systems. Thus, the genuine tripartite certifications with pure, GHZ-type, W-type, and mixed states are systematically analyzed [333] using the all-versus-nothing method [360]. The triparty correlation is experimentally verified in a photon network, which proves the existence of genuine LOSR triparty nonlocality with the inequality of Equation (32), and the locality loophole is closed [334].(32)$$F∶=I_{Bell}^{C_{1}=1}+\frac{4I_{same}-8}{1+\langle C_{1}\rangle }\leq 2\;I_{Bell}^{C_{1}=1}∶={\langle A_{0}B_{0}\rangle }_{C_{1}=1}+{\langle A_{0}B_{1}\rangle }_{C_{1}=1}+{\langle A_{1}B_{0}\rangle }_{C_{1}=1}-{\langle A_{1}B_{1}\rangle }_{C_{1}=1}\;I_{same}∶=\langle A_{0}B_{2}\rangle +\langle B_{2}C_{0}\rangle$$

When measurement independence cannot be met, which is likely to occur during actual measurement, quantifying the impact of measurement dependence on genuine nonlocality testing can improve the security of the device-independent system. The measurement dependence of the bipartite Bell test (CHSH) is expanded to the three-party Svetlichky test [310], which reveals the ability boundary of the eavesdropper (Eve) to simulate the quantum nonlocality using classical systems and provides the critical condition that genuine three-party nonlocality does not exist when measurement dependence exists in Equation (33).
7*P*_up_ + *P*_low_ > 1 or 7*P*_up_ + *P*_low_ < 1(33)

Here, *P* is the degree of measurement dependence, and *P*_up_ (*P*_low_) is the max (min) dependence in the tripartite Svetlichny test.

The assumption of independent and identically distributed (i.i.d) is a prerequisite for many device-independent protocols [337], which should be certified to close the pre-assumption loophole. The method for i.i.d certification proposed in [338] can accelerate the process and improve the accuracy by using Martingale-based protocol [361] and PBR protocol [362]. For example, only 10 experiments are needed to verify the negative lower bound of 90% with 99% confidence.

Based on the research of multipartite nonlocality within the Svetlichny scenario [231], the minimum requirement for detector efficiency is quantified for nonlocality testing under different causal constraints, which are called T_2_ local and Svetlichny-type genuine nonlocal, and the detection efficiency loophole is partially closed [363]. The efficiency of each detector must be higher than 75% to violate the three-party locality in the T_2_ scenario with non-ideal detectors, and the tolerance of background noise can reach 1.6%. In the Svetlichny-type genuine nonlocality scenario, the lower bound of detection efficiency drops to 88.1%, significantly lower than that of 97% in [364].

The monogamy relation may only describe the entanglement between two parties in the previous research [365], which cannot meet the demand for multi-party entanglement in the multipartite scenarios and may introduce loopholes. The verification of monogamy relation is extended from bipartite entanglement to multi-party scenarios based on the results in [366] through new inequalities (Equation (6) mentioned above), which can verify the information leakage in device-independent QKD networks using the inequality shown in Equation (34), which is the upper bound of the predictive power of an eavesdropper (Eve) to obtain the outcomes from legitimate parties [184]. The asymptotic secret-key rate of multipartite DI-QKD systems is also derived.(34)$$D_{Eve}\leq 2n-\frac{{\bar{\omega }}_{q}-2\sqrt{2}\left(\alpha -\left\lfloor \alpha \right\rfloor \right)}{2\left\lfloor \alpha \right\rfloor }\;D_{Eve}=D\left(\prod_{i=1}^{n}P\left(e_{i}|a_{i};x,z_{i}\right),\prod_{i=1}^{n}P\left(e_{i}|z_{i}\right)\right)$$

Here, ${\bar{\omega }}_{q}$ is defined in Equation (6), *α* = (2*^n^* − 1)/*n*, *x* is the input of the *n*-partite network, and *e_i_* and *z_i_* denote the outcome and input of Eve for recovering the output *a_i_* of the *i*-th node.

There are several hotspots in research of genuine nonlocality that can effectively support DI-QKD. The survivability of genuine network nonlocality in different noisy environments with different states is quite tricky [184,337,367,368]. The test methods of genuine nonlocality need to be optimized and combined with neural networks to simplify protocols and improve efficiency [369,370,371,372,373,374,375]. Genuine multipartite entanglement, which is the key resource of DI-QKD, needs to be constructed by simplifying the complexity of high-dimensional state manipulation with universal theoretical models and novel analytical methods [376,377,378,379].

Finally, the development of DI-QKD is briefly summarized. The DI-QKD is the ultimate solution for the security and transmission of keys [358]. But the realization of DI-QKD is very tough because of the difficulty in maintaining high-quality entangled states between remote locations with high detection efficiency [126]. Several proof-of-principle experiments on the DI-QKD have been reported [126,326,332,380,381,382]. The key transmission distance has increased from back-to-back [326,380,381] to several hundred meters [126,332] and then to several tens of kilometers [382] by using quantum random number generation [383] and heralded entanglement [384]. To promote the transmission distance and SKR of DI-QKD systems, several modified DI-QKD experimental schemes [385,386] and tools [387,388] and simplified DI-QKD protocols have been proposed [389,390,391] and demonstrated [392,393,394].

DI-QKD will continue to focus on improving protocols [331,395,396,397,398], reducing detection thresholds [399], enhancing robustness [400], improving light source performance [387,401], optimizing simplified versions of protocols [370,402,403], optimizing quantum random number generators [404,405,406,407], and designing new experiments [408].

## 5. Outlook

In the last 3 years, significant achievements and breakthroughs have been made in the research of QKD and network nonlocality, but there is still a significant gap from its widespread application, and further exploration in theory and engineering solutions is needed.

In terms of theory, significant achievements have been made in the sharing [151,164,207], certification [35,152,225], and generalization [147,181,191] of network nonlocality, especially in the proposal and verification of FNN [24,156,313] and GNN [166,234,371], the construction and verification of new inequalities [245,286,309,379], and new verification schemes [155,313], all of which provide rich quantum resources and tools for the development of QKD. The proposal of new quantum network models [184,240] provides a special basis for studying new quantum properties in networks. In terms of experiments, DI-QKD is still in the exploratory stage, and optimizing protocols [399] and random number generators [404] are currently the focus of research. The simplified version of DI-QKD will be a hot topic in future research and is likely to achieve a key transmission distance of over hundreds of kilometers [385]. MDI-QKD and TF-QKD are developing towards integration [67,94,267], long-distance [134], multi-node [129], and high key rate [67], which are also urgent requirements for engineering applications. Recently, the maximum transmission distance of satellite-based QKD has exceeded 12,900 km [54], which is a real leap towards long-distance secure quantum communication. It can be considered that the engineering application of QKD has made significant breakthroughs and is moving towards widespread applications.

## Figures and Tables

**Figure 1 entropy-27-00950-f001:**
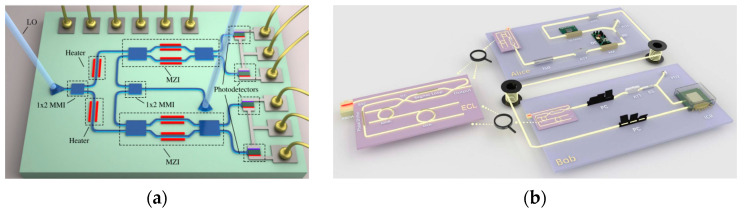
Integrated photonic transmitter and receiver used in CV-MDI-QKD systems. (**a**) Integrated receiver based on hybrid integration of silicon and GaAs with an area of 7.2 × 4.8 mm^2^. Reprinted with permission from [67]. © Optical Society of America. The chip interacts with the outside through surface grating couplers. (**b**) Integrated transmitter based on hybrid integration of Si_3_N_4_ and compounds of III-V with a footprint of 2.4 × 1.27 mm^2^. The external cavity laser adopts butterfly-shaped packaging. Reprinted with permission from [68]. © Photonics Research.

**Figure 2 entropy-27-00950-f002:**
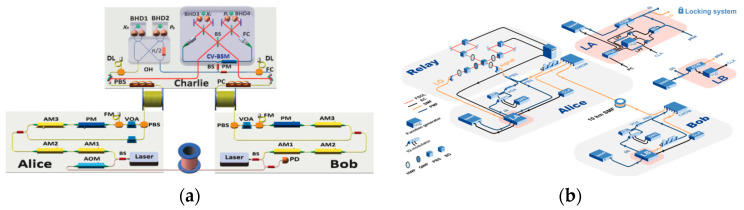
CV-MDI-QKD system with OPLL implemented by different schemes. (**a**) CV-MDI-QKD with the OPLL between Alice and Bob, which is implemented by an additional fiber channel. Reprinted with permission from [66]. © Optical Society of America. (**b**) CV-MDI-QKD with optical pilot. The pilot output by the locking system is multiplexed with quantum signals. Reprinted with permission from [70]. © IOP Publishing.

**Figure 3 entropy-27-00950-f003:**
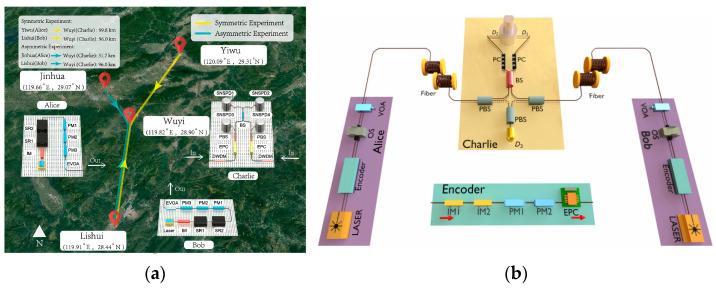
AMDI-QKD experiments with different schemes. (**a**) Field test of the AMDI-QKD system between Jinhua and Lishui and between Yiwu and Lishui with the center node located in Wuyi. Reprinted with permission from [91]. © Optical Society of America. (**b**) Lightweight AMDI-QKD without optical reference light. The duty cycle of the quantum signals reaches 100%. Reprinted with permission from [92]. © American Physical Society.

**Figure 4 entropy-27-00950-f004:**
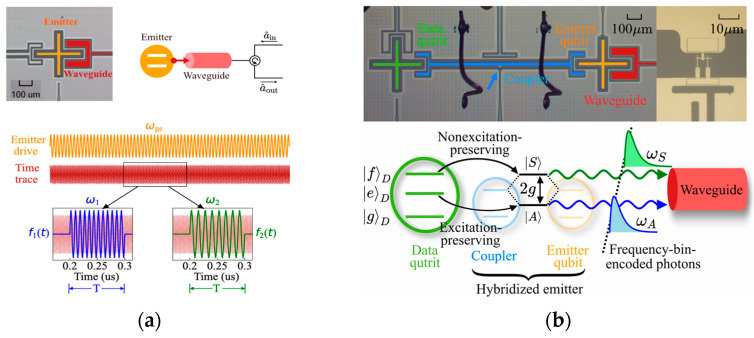
Applications of superconducting circuits and temporal mode matching in quantum computing networks. (**a**) Superconducting circuit for generating entangled photons. The temporal filters f_1_(t) and f_2_(t) are imposed on the time trace, which is the record of the radiation of the emitter, to match the input and output modes denoted by ${\hat{a}}_{in}$ and ${\hat{a}}_{out}$. Reprinted with permission from [117]. © Springer Nature. (**b**) Superconducting circuit for generating frequency-bin encoded photons. The emitter is hybridized by tuning the coupler into resonance with the qubit and emitting symmetric and anti-symmetric modes at frequencies of *ω*_S_ and *ω*_A_, respectively. Reprinted with permission from [118]. © American Physical Society.

**Figure 5 entropy-27-00950-f005:**
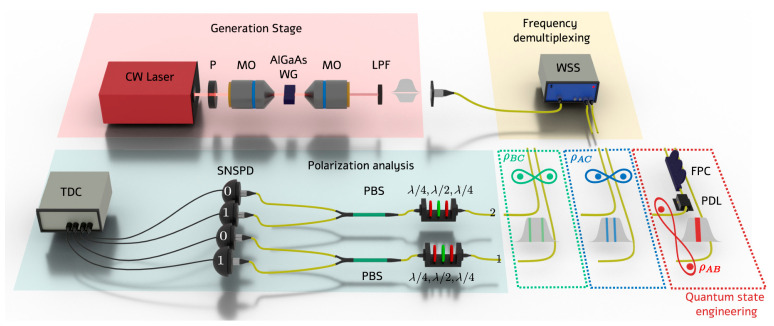
The experimental setup for the implementation of the triangle network using a single source of broadband multiplexed entangled-photon states. Reprinted with permission from [239]. © American Physical Society.

**Figure 6 entropy-27-00950-f006:**
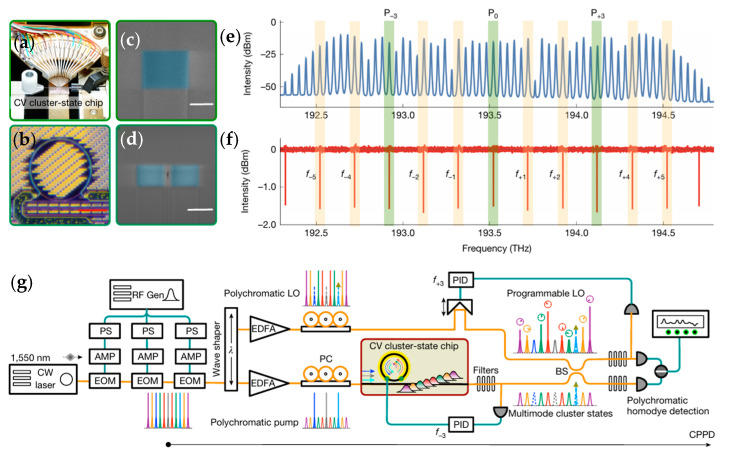
Photographs of (**a**) chip carrier, (**b**) micro-resonator, and (**c**,**d**) single-mode waveguide. (**e**,**f**) The experimental spectra and (**g**) setup for the generation and characterization of continuous-variable multi-qumode entanglement. Reprinted with permission from [279]. © Springer Nature. Lights at the frequencies of P_0,±3_ in (**e**) are selected as pumps, which are marked as the Polychromatic pump in (**g**) with three different colors. And the lights at the frequencies of f_±1,±2,±4,±5_ in (**f**) are selected as the local oscillator beams, which are marked as the Poly chromatic LO in (**g**) with different colors.

**Figure 7 entropy-27-00950-f007:**
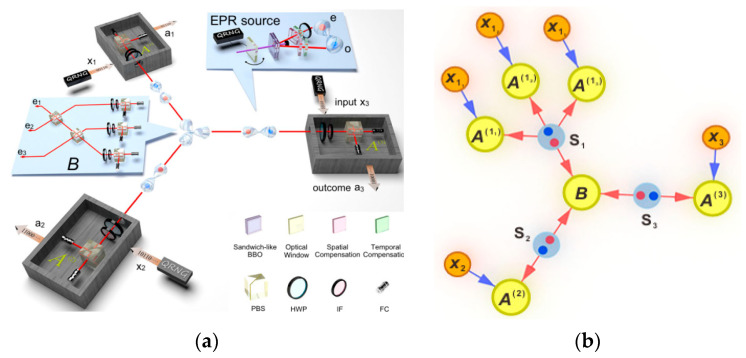
(**a**) The six-photon experimental setup to certify the FNN in a three-branch star network. (**b**) The inflation of the three-branch star network. Reprinted with permission from [299]. © Springer Nature. The red and blue dots in S_1_, S_2_ and S_3_ in (**b**) represent different polarizations, which are denoted by e- and o-light in (**a**), respectively. The blue and red arrows in (**b**) represent the electrical signals and optical signals, respectively.

**Table 1 entropy-27-00950-t001:** Several results of recent experiments on MDI-QKD systems.

Reference	Scheme	Linewidth of Laser	OFL/ Phase Locking	Security Key Rate	Detector	Security Level
[66]	CV-QKD	2 kHz and 4 kHz	OPLL/ Real time	0.43 bit/pulse @5 km (~21 kb/s) 0.19 bit/pulse @10 km (~9.5 kb/s)	APD	High (collective attack)
[67]	CV-QKD	100 Hz	Intradyne detection/ Real time	0.093 bit/symbol @5 km with 64 QAM (0.746 Gb/s) 0.019 bit/symbol @5 km with 32 QAM (0.194 Gb/s) 0.035 bit/symbol @10 km with 16 QAM (0.351 Gb/s)	Integrated PD	Very high (multiple attacks)
[70]	CV-QKD	100 Hz	OPLL/ Real time	0.13 bit/symbol @10 km (2.6 Mbit/s)	BD	High (collective attack)
[73]	FP-TF-QKD	0.1 kHz and 2 kHz	OPLL/ Real-time	8.75 × 10^−12^ bit/pulse @833.8 km (0.014 bit/s)	SSPD	Very high (decoy state)
[74]	SNS-AOPP-TF-QKD	US	OPLL/ Post-processing	3.11 × 10^−12^ bit/pulse @1002 km (0.0011 bit/s)	SNSPD	Very high (decoy state and coherent attack)
[79]	SNS-AOPP-TF-QKD	US	Not needed/ Real-time	6.4 × 10^−10^ bit/pulse @615.6 km (0.32 bit/s)	SNSPD	Very high (decoy state)
[80]	SNS-AOPP-TF-QKD	OFC	Not needed/ Real time	1.06 × 10^−9^ bit/symbol @546 km (0.53 bit/s)	SNSPD	Very high (decoy state)
[81]	SNS-AOPP-TF-QKD	~100 Hz	OIL/ Real time	2.2 × 10^−7^ bit/pulse @254 km (110 bit/s)	APD	Very high (decoy state)
[82]	NPL-TF-QKD	5 kHz	Not needed/ Post-selection	6.65 × 10^−9^ bit/pulse @504 km (2.05 bit/s)	SNSPD	Very high (decoy state)
[86]	SNS-AOPP-TF-QKD	~100 Hz	Not needed/ Post-processing	9.67 × 10^−8^ bit/pulse @502 km (~10 bit/s)	SNSPD	Very high (decoy state)
[92]	AMDI-QKD	US	Not needed/ Post-processing	6.015 × 10^−8^ bit/pulse @504.67 km (150.4 bit/s)	SNSPD	Very high (decoy state)
		100 Hz		4.004 × 10^−5^ bit/pulse @201.88 km (101.1 kbit/s)		

OFL: optical frequency locking; US: ultra-stable; SNS: sending-or-not-sending; FP: four phase; NPL: no-phase-locking; BD: balanced detectors; SSPD: superconducting single-photon detector.

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
