# Peer review of "Advances of Quantum Key Distribution and Network Nonlocality"

_entropy, 2025, doi:10.3390/e27090950_

Round 1

Reviewer 1 Report

Comments and Suggestions for Authors

    The author summarizes and analyzes the research results on QKD and network nonlocality in recent years in detail. This is helpful to identify the current status and future development paths of the QKD and network nonlocality. Therefore, I consider this paper is interesting. However, there are still some points that are suggested to be addressed for the improvement.

  1. Some symbols in the equations lack necessary explanations, such as Eqs. (1), (2), (6).
  2. In Section 3 and Section 4, the author respectively summarized the progress of quantum key distribution and network nonlocality. It is suggested to clearly explain their intrinsic connection to enhance logical coherence.
  3. Some figures are too blurry to see clearly, for example Fig. 3 (a).
  4. There are over 450 references, which is excessive. It is recommended to remove some reference that have relatively weak relevance to this manuscript.

Reviewer 2 Report

Comments and Suggestions for Authors

In this work, the authors provide an overview of the rapidly developing field of quantum
network technology. They summarize and analyze recent research findings on quantum key
distribution (QKD) and network nonlocality, including CV-MDI-QKD (continuous-variable
measurement-device-independent QKD), TF-QKD (two-field QKD), AMDI-QKD
(asynchronous MDI-QKD), the generalization, sharing, and certification of network nonlocality,
as well as the main achievements and related research tools of full network nonlocality and
genuine network nonlocality. However, before I recommend publication, the authors need to
address the following questions:

1. Some equations in the article lack explanation. For example, in Equations 1 and 2, the
specific meanings of the symbols are not explained. Although the authors cite relevant
literature, the lack of explanation of the symbols increases the difficulty of reading.
2. In the section titled “3. Quantum key distribution,” the author introduces three schemes:
CV-MDI-QKD, TF-QKD, and AMDI-QKD. However, as a summary article, it would be
more readable to explain the two types of QKD protocols — discrete-variable and
continuous-variable protocols — before introducing the specific schemes, and then
introduce the specific schemes according to their classification.
3. Some of the figures and tables in the article are too large, such as Figure 1, Figure 2, Table
1, etc.
4. The article titles need to be further verified, such as 3.3. AMDI-QKD and 3.3. Discussion.
